# Effect of Foliar Fertigation of Chitosan Nanoparticles on Cadmium Accumulation and Toxicity in *Solanum lycopersicum*

**DOI:** 10.3390/biology10070666

**Published:** 2021-07-14

**Authors:** Mohammad Faizan, Vishnu D. Rajput, Abdulaziz Abdullah Al-Khuraif, Mohammed Arshad, Tatiana Minkina, Svetlana Sushkova, Fangyuan Yu

**Affiliations:** 1Collaborative Innovation Centre of Sustainable Forestry in Southern China, College of Forest Science, Nanjing Forestry University, Nanjing 210037, China; 2Academy of Biology and Biotechnology, Southern Federal University, 344006 Rostov-on-Don, Russia; tminkina@mail.ru (T.M.); terra_rossa@mail.ru (S.S.); 3Dental Biomaterials Research Chair, Dental Health Department, College of Applied Medical Sciences, King Saud University, Riyadh 11433, Saudi Arabia; aalkhuraif@ksu.edu.sa (A.A.A.-K.); marshad@ksu.edu.sa (M.A.)

**Keywords:** agricultural production, chitosan nanoparticles, chlorophyll fluorescence, food chain, net photosynthetic rate, *Solanum lycopersicum*

## Abstract

**Simple Summary:**

The experiment conducted on *Solanum lycopersicum* provided an insight about Cd uptake, and the way a *Solanum lycopersicum* changes its physiological, biochemical and morphological responses when CTS-NPs are administered against Cd. As an effective important polymer, CTS-NPs enhanced the plant biomass, SPAD index, photosynthetic rate, and protein content in the *Solanum lycopersicum* plants grown in Cd stress, as a study herein. Addition of CTS-NPs reduced Cd accumulation by increasing the nutrient uptake. Furthermore, CTS-NPs treatment enhances tolerance to Cd stress through hampering ROS production accompanied by H_2_O_2_ activity, through reducing the peroxidation of lipids by minimizing MDA content, and through improving enzymatic (CAT, POX, SOD), non-enzymatic (GSH and AsA), and osmoprotectants (proline) antioxidant contents that are considered as a first line of defense to protect plants from stress.

**Abstract:**

Cadmium (Cd) stress is increasing at a high pace and is polluting the agricultural land. As a result, it affects animals and the human population via entering into the food chain. The aim of this work is to evaluate the possibility of amelioration of Cd stress through chitosan nanoparticles (CTS-NPs). After 15 days of sowing (DAS), *Solanum lycopersicum* seedlings were transplanted into maintained pots (20 in number). Cadmium (0.8 mM) was providing in the soil as CdCl_2_·2.5H_2_O at the time of transplanting; however, CTS-NPs (100 µg/mL) were given through foliar spray at 25 DAS. Data procured from the present experiment suggests that Cd toxicity considerably reduces the plant morphology, chlorophyll fluorescence, in addition to photosynthetic efficiency, antioxidant enzyme activity and protein content. However, foliar application of CTS-NPs was effective in increasing the shoot dry weight (38%), net photosynthetic rate (45%) and SPAD index (40%), while a decrease in malondialdehyde (24%) and hydrogen peroxide (20%) was observed at the 30 DAS stage as compared to control plants. On behalf of the current results, it is demonstrated that foliar treatment of CTS-NPs might be an efficient approach to ameliorate the toxic effects of Cd.

## 1. Introduction

Nanotechnology is an important rising field, owing to several important functions in copious research areas [1]. Nanoparticles (NPs) have special characteristics due to their small size (less than 100 nm) which results in lofty surface areas and charges; therefore, NPs are highly reactive over their bulk scale counterparts [2,3,4]. There are a number of NPs of diverse origin which can be used for multiple agricultural applications [5,6]. Recently, these particles are playing a vital role in plant tolerance to several biotic and abiotic stresses [7,8,9,10]. There are many beneficiary roles of NPs and they are used as fertilizers [11,12,13] and pesticides [14,15], having the advantage of being efficiently absorbed by crop plants, therefore having little impact on the environment over conventional fertilizers [16]. Moreover, NPs also speed-up the functioning of antioxidant enzymes, finally diminishing the accretion of reactive oxygen species (ROS) in plant cells, ameliorating stress effects leading to a better growth and yield [17,18].

Chitosan (CTS) is a conventional eco-friendly polymer, having a vast number of functions in agricultural, biomedical and fodder businesses [19]. In agricultural services, CTS acts as an antimicrobial mediator and it can help in the uptake of nutrition and in the covering of plant organs [20,21]. Chitosan nanoparticles (CTS-NPs) are used all over the world for several purposes, owing to their biodegradability, high permeability, eco-friendliness to humans and reasonable price [22]. Chitosan NPs play an important role in drug release, gene incorporation, protein shipping, exclusion of pollution, and controlled delivery of NPK fertilizers [23,24,25]. Moreover, studies have recognized that CTS might increase crop tolerance to negative conditions, including salt, cold and heat stress [26,27]. CTS-NPs have been assessed as a powerful inducer of antioxidative enzymes. Crop plant responses to these aspects are extremely composite and engross the start of genes, indoctrinating diverse proteins [28]. Resistance genes start resulting in the accrual of varied enzymes and stress-specific metabolites. The biostimulant activity of CTS-NPs has also been reported in various studies [29,30,31]. Before the use of CTS-NPs commercially, further studies are needed.

Global industrialization and the excessive use of fertilizers has led to the large scale contamination of agricultural soils, which is a pioneering threat to human beings as well as the ecosystem [32,33]. There are more than 10 million polluted sites around the world; out of them, around 50 percent are affected by trace metals [34]. Cadmium is one of the most toxic pollutants and represents a worldwide hazardous concern [35]. In the agricultural system, Cd shows proficient root-to-shoot transduction, causing abnormalities such as nutritional shortage, reticence of chlorophyll formation, decrease in photosynthesis by troubling the enzymes concerned in the Calvin cycle [36,37], restraining the stomatal aperture by prying with the plant’s water balance, infuriating oxidative stress by changing metabolism, preventing crop growth by metabolic abnormality, and finally, limiting the plant’s growth which leads to plant death [38]. Additionally, the deposition of Cd in the human body via food chain, poses severe problems and health issues, such as a high rate of cancer (prostate and lung) and bone malformation [39]. Therefore, it is very important to conquer Cd toxicity and lessen the deposition and transportation of Cd in crop plants for standard plant growth and human security. However, to our knowledge, there have been no studies that have researched the impacts of CTS-NPs on the fight of *Solanum lycopersicum* with Cd stress.

*Solanum lycopersicum* belongs to the family Solanaceae [40] and is one of the main extensive crops as one of the most important horticultural crops in the world [41]. It is largely eaten raw or processes into paste or juice [42]. The necessities of *Solanum lycopersicum* are about 90% of the dietary Vitamin C required for human nutrition, with a lofty ascorbic acid and lycopene content [43,44]. Antioxidants (anthocyanins, lycopene and carotene) are present in large amounts in *Solanum lycopersicum*. From previous studies, it is alleged that lycopene is effective in prostate cancer prevention and protecting the skin from UV rays. In *Solanum lycopersicum*, Cd is a major dilemma that limits the growth and developmental processes significantly [36,38]. Therefore, the present experiment verifies that exogenous treatment of CTS-NPs can significantly reduce the toxic effects of Cd by up-regulating the defense mechanisms.

## 2. Materials and Methods

### 2.1. Nanoparticle

The CTS-NPs were acquired from Sigma-Aldrich (St. Louis, MO, USA). Required volume (100 µg/mL) of CTS-NPs was prepared by liquefying 1 g of CTS-NPs in 10 mL of DDW in a 10 mL flask.

### 2.2. Solanum lycopersicum Cultural and Treatment Pattern

The experimentation was conducted in a randomized complete block with 20 pots (6 inch diameter) filled with soil and manure. The pots were fertigated with urea, single superphosphate and a muriate of potash, mixed at a rate of 40, 140 and 28 mg, respectively, per kg of soil to maintain the nutritional requirement of plants. Seeds of *Solanum lycopersicum* were sown to create the nursery, and at 15 DAS seedlings were transplanted to maintained pots. The Cd (0.8 mM) was provided in the solution as CdCl_2_·2.5H_2_O at the time of transplanting. The concentration of Cd was selected on the basis of our previous study [37]. Foliar application of CTS-NPs (100 µg/mL) was given at 25 DAS. The following treatments in the experiment were the control: Cd (0.8 mM), CTS-NPs (100 µg/mL) and Cd (0.8 mM) + CTS-NPs (100 µg/mL). Plants from pots were picked up at 30 DAS to assess morpho-physiological and biochemical indices.

### 2.3. Shoot and Root Dry Weight

Plants were picked up out from the pots and washed with tap water to remove any attached soil. Dry biomass of the root and shoot was measured by drying them in an oven at 72 °C for 72 h.

### 2.4. SPAD Index Measurement (Chlorophyll Content Estimation)

The Soil Plant Analysis Development (SPAD) index was calculated using the SPAD chlorophyll meter (SPAD-502; Konica, Minolta Sensing, Inc., Sakai, Osaka, Japan).

### 2.5. Chlorophyll Fluorescence

Chlorophyll fluorescence such as photosystem efficiency (Fv/Fm), PS II quantum yield (ΦPSII), photochemical quenching (qP), and non-photochemical quenching (NPQ) were calculated through a chlorophyll fluorometer (FMS 2, Hansatech Instruments Ltd., King’s Lynn, UK).

### 2.6. Leaf Gas Exchange Characteristics

The leaf gas exchange attributes, i.e., net photosynthetic rate (P_N_), stomatal conductance (gs), internal CO_2_ concentration (Ci) and transpiration rate (E), were determined as described by Faizan et al. [45]. On 30th DAS, fully expanded uppermost intact leaves were chosen to obtain readings of P_N_, gs, Ci and E in the morning between 10:00 and 11:00 a.m. using a Portable Photosynthetic System (LI-COR, Lincoln, NE, USA).

### 2.7. Lipid Peroxidation (MDA) and Hydrogen Peroxide (H_2_O_2_) Determination

The method followed by Faizan et al. [37] was followed to determine the lipid peroxidation as expressed by MDA content. Fresh leaves were homogenized in 0.1% trichloroacetic acid (TCA) and centrifuged at 10,000× *g* for 15 min. A 20% TCA solution of 0.5% thiobarbituric acid was mixed in with the supernatant. The final mixture was warmed at 95 °C for 30 min. After cooling, the supernatant was centrifuged at 1000× *g* for 15 min at 4 °C. The absorbance of the mixture was noted at 532 nm.

The amount of H_2_O_2_ in leaves was determined by the method adopted by Faizan et al. [37]. Leaf tissues were homogenized in 10 mL cold acetone with a mortar and pestle. The homogenate was centrifuged at 5000× *g* for 15 min and the supernatant was kept. Residue was again extracted with acetone. About 1 mL of mixture was taken in a test tube and 2 mL of 17 M ammonia and 2 mL of 20% titanium chloride were mixed. The supernatant was again extracted with acetone, accompanied by an infusion of 10 mL 2 N H_2_SO_4_ to absorb it properly. The optical density was measured at 410 nm on a spectrophotometer. The amount of H_2_O_2_ in the samples was measured in relation to the standard curve adopted from the known concentration of H_2_O_2_.

### 2.8. Antioxidant Enzymes

For the determination of catalase (CAT), peroxidase (POX) and superoxide dismutase (SOD), the leaf (0.5 g) was homogenized in a 50 mM phosphate buffer (pH = 7) of 1% polyvinylpyrrolidone. These mixtures were centrifuged at 15,000× *g* for 10 min at 4 °C, and the final supernatant was used as a source for the determination of CAT, POX and SOD. For the determination of POX activity, the enzyme extract (0.1 mL) was mixed in the reaction mixture of pyrogallol, phosphate buffer (pH = 6.8) and 1% H_2_O_2_. The absorbance was measured at 420 nm on a spectrophotometer [46]. For the estimation of CAT, a mixture was prepared containing a phosphate buffer (pH = 6.8), 0.1 M H_2_O_2_ and enzyme extract (0.1 mL). H_2_SO_4_ was mixed in the reaction mixture after its incubation for 1 min at 25 °C, and was titrated against a potassium permanganate solution [46]. The SOD activity was measured by the method described by Beauchamp and Fridovich [47]. For the preparation of the reaction mixture, 50 mM phosphate buffer (pH = 7.8), 20 µM riboflavin, 75 mM NBT, 13 mM methionine and 0.1 mM ethylene diamine tetra acetic acid (EDTA) were required. The mixture was irradiated within two fluorescent light tubes for 10 min and absorbance was noted at 560 nm using a UV–visible spectrophotometer.

### 2.9. Glutathione and Ascorbic Acid

The reduced GSH content was determined according to the method described by Noctor and Foyer [48]. Fresh leaf tissue (0.5 g) was ground in sulphosalicylic acid (2.0 mL of 5%) under chilling conditions. The ground tissue was centrifuged at 10,000× *g* for 10 min. In total, 0.6 mL of phosphate buffer (100 mM, pH 7.0) and 40 mL of 5, 5-dithiobis-2-nitrobenzoic acid (DTNB) were added to 0.5 mL of supernatant. After 2 min, the absorption was read at 412 nm, and calculation was performed accordingly. The AsA content was assayed following the method described by Kampfenkel et al. [49]. The leaf (300 g) was homogenized in TCA (2 mL of 6% wt./vol) and centrifuged at 15,000× *g* for 5 min at 4 °C for 15 min. Aliquots of 200 µL of the crude extract were added to a sodium phosphate buffer (800 µL of 0.2 M). The mixture was incubated at 42 °C for 15 min. Subsequently, TCA (1 mL of 10%), H_3_PO_4_ (800 µL of 42%), 2,2′-dipyridyl (800 µL of 4%) and FeCl_3_ (400 µL of 3%) were added to the mixture. After vigorous stirring, the mixture was incubated at 42 °C for 40 min. Absorbance was read at 525 nm.

### 2.10. Leaf Protein and Proline

The method of Bradford [50] was used for the estimation of protein content in plant leaves. For this, 1 g of leaves was homogenized in a buffer containing tris-HCL (40 mM), β-mercaptoethanol (0.07%), polyvinylpyrrolidone (2%), triton X-100 (0.5%), PMSF (1 mM) and EDTA (1 mM) using a mortar and pestle and centrifuged at 20,000× *g* for 10 min. Supernatant was collected and intensity was measured with a spectrophotometer.

Proline content was measured by the method of Bates et al. [51]. For estimation, 50 mg of leaves was extracted in sulfosalicylic acid, and same amount of glacial acetic acid and ninhydrin solutions were also mixed. Sample was heated at 100 °C, to which 5 mL of toluene was added. Absorbance of the aspired layer was noted at 528 nm on a spectrophotometer.

### 2.11. Plant Cd Concentration

Plant sample was washed with tap water and dried in the incubator for 48 h at 80 °C for the determination of plant Cd content. The dried sample was weighed, ground into a fine powder using a mortar and pestle, and then mingled with a concentration of HNO_3_/HClO_4_ (4:1). The Cd concentration was measured by inductively coupled plasma optical emission spectroscopy (ICP-OES, Optima, 7000).

### 2.12. Statistical Analysis

Data were statistical assessed and standard errors (±) were calculated (n = 5). Analysis of variance (ANOVA) was carried out using SPSS (ver. 17 for windows, IBM Corporation). The least significant difference was calculated for the significant data at *p* < 0.05.

## 3. Results

### 3.1. Dry Weight

The dry weight of *Solanum lycopersicum* grown under Cd stress was decreased as compared to control; however, plants treated with CTS-NPs (100 µg/mL) showed a significant enhancement in the dry weight of shoot and root by 37.6% and 9.7% compared to the control, respectively (Figure 1A,B).

### 3.2. SPAD Index

The Cd toxicity caused a reduction in the SPAD index of *Solanum lycopersicum* in comparison to the control, but a considerable increase in chlorophyll content was observed in plants treated with 100 µg/mL of CTS-NPs grown under Cd stress (Figure 1C). In *Solanum lycopersicum,* CTS-NPs showed an increment in the SPAD index of 40.2% as compared to control plants (Figure 1C).

### 3.3. Chlorophyll Fluorescence

Chitosan NPs increased Fv/Fm, ΦPSII and qP (Figure 1D–F). However, CTS-NPs treatment showed a decline in NPQ (Figure 1G). When CTS-NPs were used, the reduction in NPQ was 35% over the control. The Cd stress decreased Fv/Fm by 33%, ΦPSII by 23.3% and qp by 25.7% over the water treated plant, but increased NPQ by 42%. The fertigation of CTS-NPs in Cd-contained plants caused a rise in Fv/Fm and ΦPSII of about 65% and 40%, an increase in qP of 48%, and a decrease in NPQ of 28%, compared to only Cd-treated plants.

### 3.4. Leaf Gas Exchange Characteristics

The application of CTS-NPs only or in combination with Cd-stressed plants augmented the P_N_, gs, Ci and E over untreated plants (Figure 2A–D). In contrast, Cd stress decreased P_N_, gs, Ci and E by 31.4%, 38.1%, 33.5% and 37.6%, respectively, compared to non-treated plants. The application of CTS-NPs in Cd-stressed plants enhanced P_N_, gs, Ci and E by 30.44%, 36.35%, 37% and 40.67%, respectively, compared to Cd-stressed plants (Figure 2A–D).

### 3.5. Lipid Peroxidation (MDA) and Hydrogen Peroxide (H_2_O_2_) Determination

Cadmium stress significantly increased the concentration of MDA and H_2_O_2_ by 43% and 47%, respectively, over only water-treated plants (Figure 3A,B). However, CTS-NPs decreased the concentration of MDA and H_2_O_2_ in the absence or presence of Cd stress.

### 3.6. Antioxidant Enzymes

The Cd stress increased the activities of CAT (52%), POX (65%) and SOD (39%) in contrast to the control plants (Figure 3C–E). However, foliar treatment of CTS-NPs in the absence/presence of Cd stress further increased the activity of CAT, POX and SOD. The maximum increase was noted in the plants treated with CTS-NPs + Cd by 71%, 79% and 44% in CAT, POX and SOD, respectively, over the control plants (Figure 3C–F).

### 3.7. Glutathione and Ascorbic Acid

The addition of Cd in the soil increased the concentrations of GSH (39%) and AsA (61%) over their control (Figure 4A,B). Moreover, the addition of CTS-NPs in the Cd-stressed plant further increased the concentrations of GSH and AsA, which were 45% and 66% higher than the non-treated plants (Figure 4A,B).

### 3.8. Leaf Protein and Proline

Leaf protein content significantly decreased through the exposure of Cd stress by 30% over control plants (Figure 4D). However, the foliar application of CTS-NPs increased the protein content by 17% compared to the control plants (Figure 4D).

The leaf proline content was significantly increased (17%) in the presence of Cd; however, the foliar application of CTS-NPs showed a marked reduction in the proline content over the water-treated control plants (Figure 3F).

### 3.9. Plant Cd Concentration

The Cd concentration significantly enhanced in the plants subjected with 0.8 mM of Cd through soil. However, the foliar application of CTS-NPs drastically decreased the Cd concentration in *Solanum lycopersicum* plants (Figure 4C).

## 4. Discussion

Nanotechnology has the capability to grant a new technology-based agricultural revolution [52]. The involvement of polymeric NPs in agriculture is frequently rising at present because of their biocompatibility, reproducibility and capability to answer to outer stimuli [53]. Moreover, CTS is an accepted non-hazardous polymer that has been employed as a plant growth stimulator [20]. Latest research has documented that CTS persuades the mechanism in plants against the several biotic and abiotic stresses [54] and aids in the arrangement of stoppers that enhance crop yield [55].

The toxicity of Cd caused a reduction in cell division and cell elongation, ultimately limiting the plant’s growth and development [56,57]. The exogenous appliance of CTS-NPs significantly increased the growth of *Solanum lycopersicum* as confirmed through the boost in dry mass. This designates that CTS-NPs enhanced the growth of *Solanum lycopersicum*. Stomata provide a more proficient way for NP uptake by an elevated stomata density principally to promote a faster and higher uptake of NPs [58,59]. Along this, NPs also enter via the bases of the trichomes, cuticle and epidermis of leaves [60,61]. The present results are lined with other studies where CTS mitigated the adverse impacts of Cd stress on *Flamingo anthurium* [62] and *Brassica napus* [63].

The chlorophyll content is a vital indicator of Cd stress tolerance in plants. In the current observation, Cd stress reduced the SPAD index in leaves of *Solanum lycopersicum* (Figure 1C). It was observed that CTS, as a penetrant particle, was able to pass through the leaf via the stomata and play a significant function in transporting water into the plant [64]. Limpanavech et al.’s [65] study suggested that the foliar application of CTS-NPs onto leaves of a *Dendrobium* orchid could decrease *ycf2* gene expression and expand the chloroplast size in leaves. The current results are agreement with the previous study of Pirbalouti et al. [66] on *Ocimum basilicum*. In comparison with the current results, the impacts of CTS in escalating chlorophyll amounts were established in *Cucumis sativus*, *Raphanus sativus* and *Vigna unguiculata* [67,68].

The chlorophyll fluorescence procedure was confirmed to be a receptive procedure for the recognition and evaluation of alternate methods encouraged in the photosynthetic machinery. The Fv/Fm, Fv/F0, qP and NPQ assessments were also employed to conclude CTS-NP-triggered amelioration in the photosynthetic machinery. The presence of Cd in the soil reduced the values of Fv/Fm, Fv/F0 and NPQ; however, the foliar application of CTS-NPs on Cd-treated *Solanum lycopersicum* mitigated these adverse effects (Figure 1D–G). These observations show that CTS-NPs could upgrade fluorescence pigments and perform defensive impacts on the photosynthetic machinery of *Solanum lycopersicum* exposed to Cd stress. Apart from this, the NPQ value considerably improved under Cd stress, showing that the antenna pigment was unable to convert light energy into chemical energy, therefore, free as high temperature [54].

The gaseous exchange characteristic provided an insight into the plant’s physiological adaptation under Cd stress [37,69]. In the current study, the foliar application of CTS-NPs increased the activity of leaf gas exchange parameters (Figure 2A–D). Several reports suggest that CTS application significantly increases P_N_, gs, Ci and E in Cd-stressed plants [70,71,72]. Liu et al. [72] reported that CTS-NPs improved P_N_, gs, Ci and E in Cd-stressed *Triticum aestivum*. Some reports suggest that CTS application enhances photosynthesis and nutrient assimilation in abiotic stress attributing to high gaseous exchange and a reduced Na^+^ assimilation by diminishing E [71]. The treatment of CTS improved photosynthesis by a raise in photosynthetic pigments, photosynthetic features, augmentation in water potential, lessening in oxidative stress [72], and mitigation of chloroplast damage [73], stomatal limitations [71], chlorophyll content [70], and an increase in RuBisCo activity [74]. The application of CTS-NPs increased the activity of gs and E, which is due to an increased water assimilation in response to water loss and it maintains water equilibrium even under abiotic stress conditions [71].

MDA is an important product of cell membrane lipid peroxidation and it can be used to measure the degree of cell membrane damage. In the present study, Cd stress significantly increased the MDA level in *Solanum lycopersicum* (Figure 3A). However, the foliar application of CTS-NPs reduced the MDA accumulation in Cd-stressed *Solanum lycopersicum* plants, suggesting that CTS-NPs have the capability to reduced lipid peroxidation. Results were in conformity with the studies of Zou et al. [54] in *T. aestivum* and Yang et al. [75] in *Malus domestica*, where CTS significantly reduced the level of MDA under salt and drought stress, respectively. The observation demonstrated that CTS may diminish the negative responses of ROS for membranes and lower the accumulation of H_2_O_2_, O_2_ and HO, etc., maybe via stimulating the ROS forage enzymes. It has been reported that Cd stress might cause oxidative stress by generating excess ROS (Figure 5) [76]. Based on the present results, it was demonstrated that antioxidative enzyme activity significantly increased by the exogenous application of CTS-NPs (Figure 3C–E and Figure 5). CAT, POX and SOD are important enzymes in plants [77], and provide the primary procession of protection adjacent to ROS by catalyzing the dismutation of O_2_ into H_2_O_2_ [78]. Peroxidase catalyzes the H_2_O_2_-reliant oxidation of a substrate, while CAT converts H_2_O_2_ to H_2_O and O_2_ [79]. The combined action of CAT, POX and SOD can limit the adverse effects caused by MDA to the cell membrane [80]. Similarly, Faizan et al. [37,45] reported that ZnO-NP addition improved the activities of antioxidant enzymes such as CAT, POX and SOD, but decreased the level of MDA and H_2_O_2_ in *Solanum lycopersicum* under Cu and Cd stress, respectively.

AsA-GSH plays a significant role as an indirect performer of the antioxidant system in plants. The current study suggested that the foliar application of CTS-NPs in the presence/absence of Cd-stressed plants increased the level of AsA and GSH (Figure 4A,B). A higher glutathione reductase activity helped to maintain the reduced GSH accumulation and higher redox state. The dehydroascorbate reductase enzyme uses GSH as the substrate to reduce DHA to AsA; GSH was oxidized to GSSG [81,82]. It was glutathione reductase that performed the role in reducing the generated GSSG back to GSH by an NADPH-dependent reaction [83].

Proline is a major organic solute that assists in cell osmoregulation under abiotic stress conditions. In the present study, it was observed that the proline content was increased in Cd-stressed plants; however, the foliar application of CTS-NPs reduced the proline content in *Solanum lycopersicum* plants under Cd stress (Figure 3F). It was previously proved that proline accretion is a latent indicator of stress patience [56,79]. Moreover, an accretion of proline is supposed to keep plant tissues safe from osmotic stress by accrue fluids well-matched with osmoregulation, the chelation and detoxification of metals, defense of enzymes, regulation of cytosolic acidity, establishment of the apparatus of protein formation and trapping of ROS [84].

In the present experiment, Cd stress to *Solanum lycopersicum* was concluded in a lesser protein content (Figure 4D). This might be due to a decrease in protein synthesis under Cd stress and the degradation of protein by protease activity, leading to an enhanced degree of protein denaturation [85]. The CTS-NPs application significantly increased the protein content in *Solanum lycopersicum* under Cd stress (Figure 4D). The negatively charged phosphate groups of nucleic acids can strongly interact with chitosan. This direct attachment can stimulate particular modifications in the expression and activity of proteins involved in the stress response [86]. The present results are lined with the previous studies on *Curcuma longa*, *Prunus davidiana* and *V. radiata* [87,88,89].

## 5. Conclusions

It is concluded from the present work that CTS-NPs application successfully enhanced growth, photosynthesis, protein content and antioxidant enzymes under Cd stress conditions, which may be activated by several biochemical and physiological mechanisms. The foliar application of CTS-NPs also limited MDA, H_2_O_2_ and proline content and Cd concentration in *Solanum lycopersicum*. Furthermore, CTS-NPs could probably lift efficiently sustainable agriculture not only regarding *Solanum lycopersicum,* but also in other crops in the future as an evolving methodology employing nanotechnology and agriculture.

## Figures and Tables

**Figure 1 biology-10-00666-f001:**
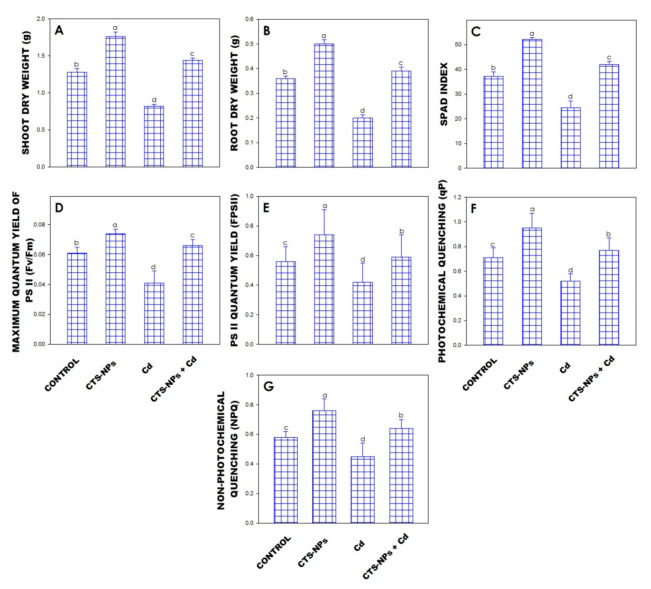
Effect of foliar application of chitosan nanoparticles (CTS-NPs) on shoot dry weight (**A**), root dry weight (**B**), SPAD index (**C**), maximum quantum yield of PSII (**D**), PSII quantum yield (**E**), photochemical quenching (**F**) and non-photochemical quenching (**G**) in cadmium (Cd) stress *Solanum lycopersicum* at 30 days after sowing. Values are the mean of five replicates with standard errors. Different bar letters show significant differences among treatments separately.

**Figure 2 biology-10-00666-f002:**
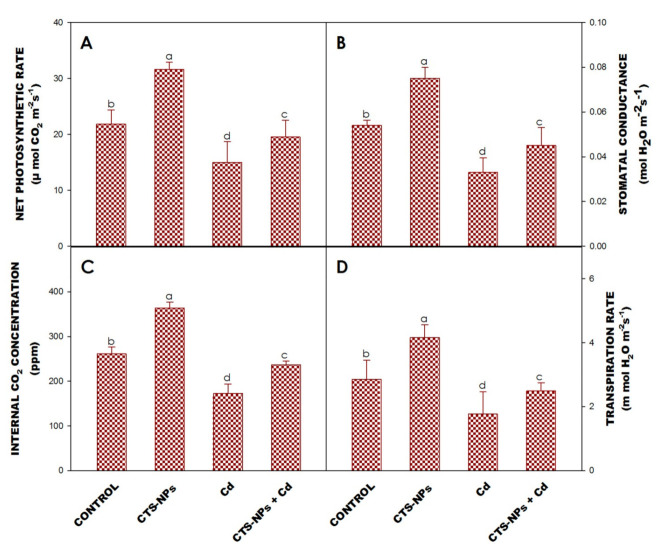
Effect of foliar application of chitosan nanoparticles (CTS-NPs) on net photosynthetic rate (**A**), stomatal conductance (**B**), internal CO_2_ concentration (**C**) and transpiration rate (**D**) in cadmium (Cd) stress *Solanum lycopersicum* at 30 days after sowing. Values are the mean of five replicates with standard errors. Different bar letters show significant differences among treatments separately.

**Figure 3 biology-10-00666-f003:**
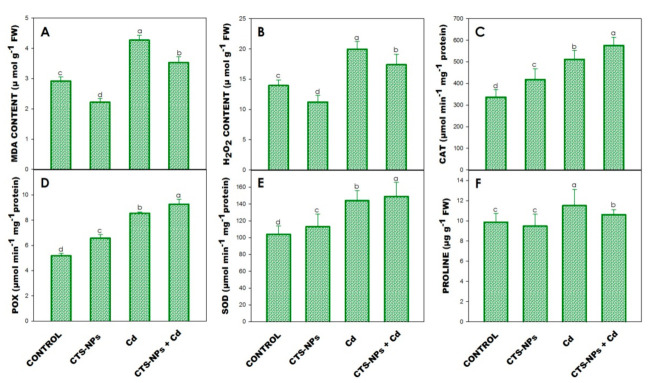
Effect of foliar application of chitosan nanoparticles (CTS-NPs) on the contents of MDA (**A**), H_2_O_2_ (**B**), CAT (**C**), POX (**D**), SOD (**E**) and proline (**F**) in cadmium (Cd) stress *Solanum lycopersicum* at 30 days after sowing. Values are the mean of five replicates with standard errors. Different bar letters show significant differences among treatments separately.

**Figure 4 biology-10-00666-f004:**
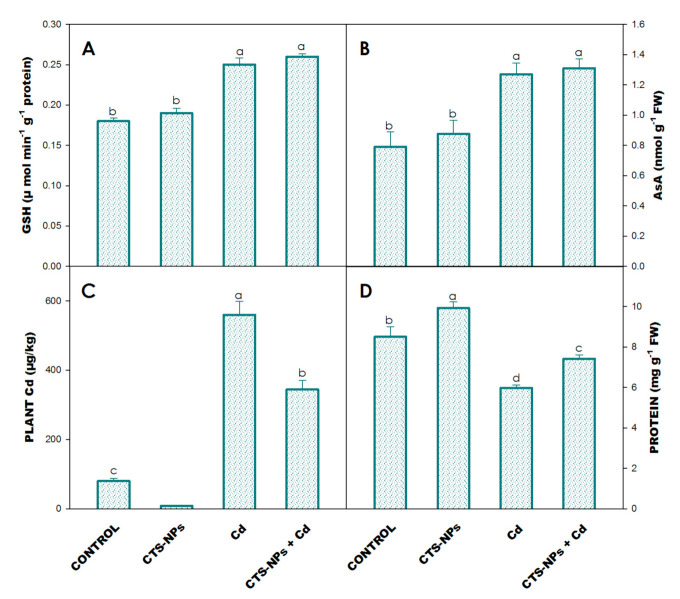
Effect of foliar application of chitosan nanoparticles (CTS-NPs) on the level of GSH (**A**), AsA (**B**), plant Cd (**C**) and protein (**D**) in cadmium (Cd) stress *Solanum lycopersicum* at 30 days after sowing. Values are the mean of five replicates with standard errors. Different bar letters show significant differences among treatments separately.

**Figure 5 biology-10-00666-f005:**
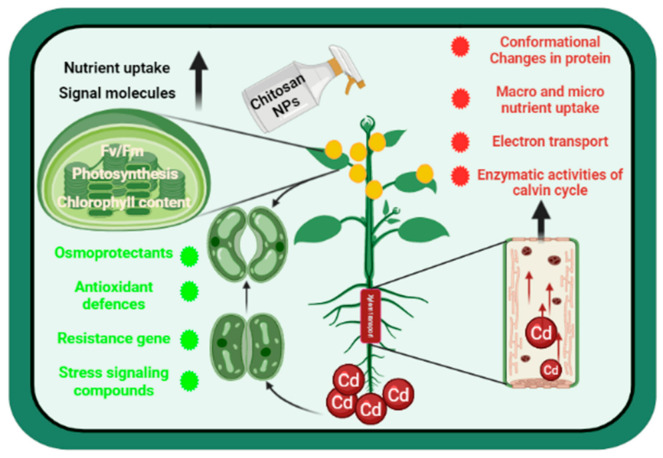
Schematic diagram shows the impact of chitosan nanoparticles to encountering the cadmium stress in plants. Red and green dots indicate decrease and increase in a parameter, respectively.

## Data Availability

Not applicable.

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
