# Peer review of "Effect of Foliar Fertigation of Chitosan Nanoparticles on Cadmium Accumulation and Toxicity in Solanum lycopersicum"

_biology, 2021, doi:10.3390/biology10070666_

Round 1

Reviewer 1 Report

“Elucidating the impact of chitosan nanoparticles under cadmium stress in Solanum lycopersicum”.

I find the paper and the concept of the research very interesting, especially in regards of the impact of the chitosan nanoparticles on the plants grown under cadmium stress. The results presented in a paper are well described, condensed and clearly show that CTS-NPs effectively reduced Cd toxicity.

I have one question. You are showing the plants dry weight during the experiment but what about the length of stems and roots or number of leaves. Does application of chitosan nanoparticles had effect on these parameters?

Overall, this paper is a valuable addition to the research of nanoparticles and their effect on plants.

Author Response

Author responses to reviewer (1) comments

(biology-1278460 R1)

To Reviewer #1

Remarks: I find the paper and the concept of the research very interesting, especially in regards of the impact of the chitosan nanoparticles on the plants grown under cadmium stress. The results presented in a paper are well described, condensed and clearly show that CTS-NPs effectively reduced Cd toxicity.

Response: The authors are very thankful to the anonymous Reviewer for the appreciation, valuable suggestions, comments and scientific criticism of manuscript for its further improvement.

Remarks: I have one question. You are showing the plants dry weight during the experiment but what about the length of stems and roots or number of leaves. Does application of chitosan nanoparticles had effect on these parameters?

Response: Yes of course length of stems and roots or number of leaves definitely affected with the application of chitosan nanoparticles, but, I had determined only plant dry weight in the present experiment.

Remarks: Overall, this paper is a valuable addition to the research of nanoparticles and their effect on plants.

Response: The reviewer comments are reasonable, and we have corrected the MS in accordance with the comments and suggestions. A thorough internal reviews was performed in the whole MS, changes highlighted in Track Change Format supplied MS. We are thankful to learned reviewer for giving critical insights, leading to substantial improvement in the manuscript, we hope the response meets the reviewer approval.

Reviewer 2 Report

I have reviewed the manuscript and please find suggestions on the manuscript of Faizan et et al. entitled " Elucidating the Impact of Chitosan Nanoparticles under Cadmium Stress in Solanum lycopersicum."

The present study beneficial effect of foliar application of Chitosan nanoparticle on  mitigating Cd toxicity. The study is well conducted supported by different morphological, physiological and bioschemical parameters. The manuscript needs to extensively revised for English. The scientific content of the manuscript is good but needs to be presented in a better way and should be addressed before publication in Biology. It needs careful reading before submission. My major concerns are listed in the manuscript itself. Please look at the comments throughout the manuscript and needs to be addressed. I will recommend substantial revision in the manuscript based on the comments listed in PDF version of manuscript.

Author Response

Author responses to reviewer (2) comments

(biology-1278460 R1)

To Reviewer #2

Remarks: I have reviewed the manuscript and please find suggestions on the manuscript of Faizan et al. entitled “Elucidating the Impact of Chitosan Nanoparticles under Cadmium Stress in Solanum lycopersicum." The present study beneficial effect of foliar application of Chitosan nanoparticle on mitigating Cd toxicity. The study is well conducted supported by different morphological, physiological and biochemical parameters. The manuscript needs to extensively revised for English. The scientific content of the manuscript is good but needs to be presented in a better way and should be addressed before publication in Biology. It needs careful reading before submission. My major concerns are listed in the manuscript itself. Please look at the comments throughout the manuscript and needs to be addressed. I will recommend substantial revision in the manuscript based on the comments listed in PDF version of manuscript.

Response: The authors are very thankful to the anonymous #Reviewer2 for the appreciation, valuable suggestions, comments and scientific criticism of manuscript for its further improvement. Suggestions/comments given in manuscript PDF proved to be very important in improving the scientific, technical and language quality of the manuscript for its better understanding to the readers. All the suggestions and comments of the have been accepted by the authors and the manuscript has been corrected accordingly. A thorough internal reviews was also performed in the whole MS for possible improvement, changes highlighted in Track Change Format supplied MS. We hope the response meets the reviewer approval.

Reviewer 3 Report

This paper presents an interesting topic related to the physiological, biochemical and morphological responses of Solanum lycopersicum plants, when they are subjected to Cd stress in the presence of Chitosan Nanoparticles.

Some clarifications are needed before it can be accepted for publication:

row 113 Plant dry weight - it is not clear if only the roots and stems were used or the leaves were also included (if Plant dry weight is mentioned, the analyzes should include the whole plant - although in Results, row 157 refers only at the stem and root).

row 117 Chlorophyll content - what / how many leaves were used to calculate the chlorophyll content?

Depending on what parts/length of the life cycle of the plant, the time of foliar application of CTS-NPs and the time of harvest were chosen?

Given that S. lycopersicum is an agricultural plant, was it not more interesting to follow the effect of CTS-NPs on Cd accumulation in fruits (or in fruits, in addition to vegetative organs)?

This is all the more necessary as the authors conclude that CTS-NPs can be used in sustainable agriculture. The effect of CTS-NPs on the accumulation of cadmium in the vegetative organs of the plant and their possible translocation into fruit should be discussed in more depth; it is observed from figure 4 that however CTS-NPs does not decrease Cd from the analyzed plants to a level comparable to that of Cd from the control plant.

row 322 - In the present lesson..- may be In the present paper.

I suggest authors to consider this latest reviews on the role of chitosan-based derivatives in plant protection against biotic and abiotic stresses:

Malerba, M., & Cerana, R. (2020). Chitin-and Chitosan-Based Derivatives in Plant Protection against Biotic and Abiotic Stresses and in Recovery of Contaminated Soil and Water. Polysaccharides1(1), 21-30.

Author Response

Author responses to reviewer (3) comments

(biology-1278460 R1)

To Reviewer #3

Remarks: This paper presents an interesting topic related to the physiological, biochemical and morphological responses of Solanum lycopersicum plants, when they are subjected to Cd stress in the presence of Chitosan Nanoparticles. Some clarifications are needed before it can be accepted for publication:

Response: The authors are very thankful to the anonymous Reviewer#3 for the appreciation, valuable suggestions, comments and scientific criticism of manuscript for its further improvement. Suggestions/comments given by the reviewers proved to be very important in improving the scientific, technical and language quality of the manuscript for its better understanding to the readers. All the suggestions and comments have been accepted by the authors and the manuscript has been corrected accordingly.

Remarks: row 113 Plant dry weight - it is not clear if only the roots and stems were used or the leaves were also included (if Plant dry weight is mentioned, the analyzes should include the whole plant - although in Results, row 157 refers only at the stem and root).

Response: In the present study only shoot and root dry weight were measured. Also correction has been done.

Remarks: row 117 Chlorophyll content - what / how many leaves were used to calculate the chlorophyll content?

Response: Five upper most leaves were used for the estimation of SPAD chlorophyll. Remarks: Depending on what parts/length of the life cycle of the plant, the time of foliar application of CTS-NPs and the time of harvest were chosen?

Response: Foliar application of CTS-NPs was done when true leaves had come, however after five days of CTS-NPs application, harvest was done because wanted to see that what the impact of CTS-NPs was in five days.

Remarks: Given that S. lycopersicum is an agricultural plant, was it not more interesting to follow the effect of CTS-NPs on Cd accumulation in fruits (or in fruits, in addition to vegetative organs)?

Response: Yes, it is an agricultural plant, fruit yield will be calculated at 180 days after sowing, and if we calculate the Cd accumulation in fruits, it will take the same time. But we have the limited time to complete this experiment. In the next experiment we will consider it.

Remarks: This is all the more necessary as the authors conclude that CTS-NPs can be used in sustainable agriculture. The effect of CTS-NPs on the accumulation of cadmium in the vegetative organs of the plant and their possible translocation into fruit should be discussed in more depth; it is observed from figure 4 that however CTS-NPs does not decrease Cd from the analyzed plants to a level comparable to that of Cd from the control plant.

Response: I will plan the next experiment to carry out the omics/proteomic analysis of tomato fruits and also try to recognize the gene responsible for tolerance against Cd after application of CTS-NPs. However, according to figure 4, CTS-NPs decreased the Cd concentration in plants but not equal to control. The objective of this experiment was to check, is CTS-NPs applicable to lower down the adverse effects of Cd and it is proved in the results that CTS-NPs mitigate the toxicity caused by Cd.

Remarks: row 322 - In the present lesson..- may be In the present paper.

Response: Corrected as per your suggestion.

Remarks: I suggest authors to consider this latest reviews on the role of chitosan-based derivatives in plant protection against biotic and abiotic stresses:

Remarks: Malerba, M., & Cerana, R. (2020). Chitin-and Chitosan-Based Derivatives in Plant Protection against Biotic and Abiotic Stresses and in Recovery of Contaminated Soil and Water. Polysaccharides1(1), 21-30.

Response: Reference added as per your suggestion.

A thorough internal reviews was also performed in the whole MS, changes highlighted in Track Change Format supplied MS. We are thankful to learned reviewer for giving critical insights, leading to substantial improvement in the manuscript, we hope the response meets the reviewer approval.

Reviewer 4 Report

Review

of

Elucidating the Impact of Chitosan Nanoparticles under Cadmium Stress in Solanum lycopersicum

Keywords: ’Agricultural production, Chlorophyll fluorescence, Food chain, Net photosynthetic  rate ’

Please include Chitosan Nanoparticles  and Solanum lycopersicum into the keywords

line 117 ’ Chlorophyll content’   

SPAD index is not chlorophyll content,  it is an estimation of chlorophyll content! Please change the title and its reference in the text.

2.7-2.11

Please give detailed information on sample preparation and measurement (solvents, equippments etc.), not only references!

Please explain, how is it possible that the plant Cd content was lower when Cd enters the plants via roots and CNPs were foliar applied?

Do you have any evidence of xylem transport towards the root region?

line 257 What do you mean by ’ Cd-triggered amelioration to the photosynthetic machinery’?

Author Response

Author responses to reviewer (4) comments

(biology-1278460 R1)

To Reviewer #4

Remarks: Keywords: ’Agricultural production, Chlorophyll fluorescence, Food chain, Net photosynthetic rate ’

Please include Chitosan Nanoparticles and Solanum lycopersicum into the keywords

Response: Added as per your suggestion.

Remarks: line 117 Chlorophyll content’ SPAD index is not chlorophyll content; it is an estimation of chlorophyll content! Please change the title and its reference in the text.

Response: Corrected as per your suggestion

Remarks: 2.7-2.11- Please give detailed information on sample preparation and measurement (solvents, equippments etc.), not only references!

Response: Detailed information of the material and method used has been described as per your suggestion.

Remarks: Please explain, how is it possible that the plant Cd content was lower when Cd enters the plants via roots and CNPs were foliar applied?

Response: In the experiment, Cd concentration was measured in the leaf, shoot and root separately. The data presented in the manuscript is the mean of Cd concentration of all the separated plant parts. It is clearly observed that after chitosan nanoparticle application Cd content reduced in the plant.

Remarks: Do you have any evidence of xylem transport towards the root region?

Response: No

Remarks: line 257 what do you mean by’ Cd-triggered amelioration to the photosynthetic machinery’?

Response:  It was the writing mistake; now correct it “Cd” in “CTS-NPs”.

A thorough internal reviews was also performed in the whole MS, changes highlighted in Track Change Format supplied MS. We are thankful to learned reviewer for giving critical insights, leading to substantial improvement in the manuscript, we hope the response meets the reviewer approval.

Reviewer 5 Report

I am interested in the results of the study entitled Elucidating the Impact of Chitosan Nanoparticles under Cadmium Stress in Solanum lycopersicum. I would like to inform you that this paper can be printed for the following reasons.

For CD stress in S. lycopersicum plants, the results of CTS-NPs increasing plant biomass, chlorophyll content, photosynthetic rate, and protein content are very good studies. However, it would be better if the clarity of all figures was increased.

Author Response

Author responses to reviewer (5) comments

(biology-1278460 R1)

To Reviewer #5

Remarks: I am interested in the results of the study entitled “Elucidating the Impact of Chitosan Nanoparticles under Cadmium Stress in Solanum lycopersicum”. I would like to inform you that this paper can be printed for the following reasons.

Response: The authors are very thankful to the anonymous Reviewer for the appreciation, valuable suggestions, comments and scientific criticism of manuscript for its further improvement.

Remarks: For CD stress in S. lycopersicum plants, the results of CTS-NPs increasing plant biomass, chlorophyll content, photosynthetic rate, and protein content are very good studies. However, it would be better if the clarity of all figures was increased.

Response: High quality figures have been inserted as per your suggestion.

A thorough internal reviews was performed in the whole MS, changes highlighted in Track Change Format supplied MS. We are thankful to learned reviewer for giving critical insights, leading to substantial improvement in the manuscript, we hope the response meets the reviewer approval.

Round 2

Reviewer 4 Report

Dear Authors,

thank you for the work and the efforts you have put into the improvement of the MS.

2.4. Chlorophyll contentSPAD chlorophyll Chlorophyll content The Soil Plant Analysis Development (SPAD) chlorophyll was calculated using SPAD chlorophyll meter (SPAD- 502; Konica, Minolta sensing, Inc., Japan). 

Please change it to SPAD index measurement (chlorophyll content estimation)

In several cases, Solanum. lycopersicum is written with a dot between the two word. Please correct!

Author Response

Author responses to reviewer (4) re-evaluating comments

(biology-1278460 R1)

To Reviewer #4

Remarks: thank you for the work and the efforts you have put into the improvement of the MS.

Response: The authors are very thankful to the anonymous Reviewer for the re-evaluating the manuscript and their appreciation, valuable suggestions, comments and scientific criticism of manuscript for its further improvement.

Remarks: 2.4. Chlorophyll content SPAD chlorophyll Chlorophyll content The Soil Plant Analysis Development (SPAD) chlorophyll was calculated using SPAD chlorophyll meter (SPAD- 502; Konica, Minolta sensing, Inc., Japan). 

Please change it to SPAD index measurement (chlorophyll content estimation)

Response: Correction has been made in the original manuscript in Track Change Manner.

Remarks: In several cases, Solanum. lycopersicum is written with a dot between the two word. Please correct!

Response: Manuscript revised very carefully and correction has been corrected.

I express my deep thanks to the reviewer. We hope the new manuscript will meet your journal’s standard.
